# Strengthening Peer Mentoring Relationships for New Mothers: A Qualitative Analysis

**DOI:** 10.3390/jcm11206009

**Published:** 2022-10-12

**Authors:** Kwok Hong Law, Ben Jackson, Xuan Hui Tan, Samantha Teague, Amanda Krause, Kaila Putter, Monique Du’cane, Lisa Gibson, Kelby F. Bulles, Jennifer Barkin, James A. Dimmock

**Affiliations:** 1College of Healthcare Sciences, James Cook University, Townsville, QLD 4811, Australia; 2Telethon Kids Institute, Perth, WA 6009, Australia; 3School of Human Sciences (Exercise and Sports Science), University of Western Australia, Perth, WA 6009, Australia; 4Cairnmillar Institute, Hawthorn East, VIC 3123, Australia; 5School of Medicine, Mercer University, Macon, GA 31207, USA

**Keywords:** Australia, perinatal, stress, postpartum, thematic analysis

## Abstract

(1) Background: The transition to motherhood can be challenging, especially for first-time mothers, and can accompany maternal distress. Social support—such as that offered by peers—can be important in assisting mothers to manage such distress. Although primiparous mothers often seek out and value peer support programs, few researchers have investigated factors that may influence the strength of relationships in non-professional maternal peer support programs. Insight into these factors can be key to enhancing the success of future peer support interventions. (2) Methods: Reflexive thematic analysis was applied to data gathered from 36 semi-structured interviews conducted with 14 primiparous mothers and 17 peer mentors in a peer support program. (3) Results: Four themes related to successful mentorship were identified: expectations of peer relationship, independence of peer mentor, contact, and similarities. (4) Conclusions: For primiparous mothers who are developing their support network, these factors appear important for promoting close and effective peer support relationships. Interventions that harness the dynamics between these factors may contribute to more successful peer support relationships and mental health outcomes for participants.

## 1. Introduction

The perinatal period is characterised by significant changes in women’s careers, identities, routines, and relationships [1,2]. Alongside substantial physical and physiological changes during this period, social and emotional challenges place mothers at significant risk of postpartum mental illness. Even before the recent pandemic, the global population prevalence of self-reported postnatal depression was 18% [3], and self-reported perinatal anxiety affected 18.2% of mothers [4]. A recent meta-analysis revealed that the prevalence of postpartum depression increased significantly since the start of COVID-19, with a reported pooled prevalence—across multiple data sets from multiple nations—of 34% [5]. As well as causing significant suffering for women, postnatal depression and anxiety can have downstream effects on birth and neonatal outcomes [6,7], mother-infant interactions [8], children’s cognitive, language, motor, social-emotional and adaptive behaviour development [9], and partner mental health [10]. Given the significant adverse implications of mothers’ experiences of postnatal depression and anxiety, researchers have sought to investigate factors that may improve perinatal mental health. One of these factors is social support, the absence of which is regarded as one of the strongest predictors of postpartum depression [11,12].

Social support refers to relationships and interactions that can improve a person’s mental or physical health [13]. Conceptually and operationally, social support is often separated into four different aspects: (1) informational support—providing information to individuals to deal with their existing problems, (2) emotional support—showing care, concern, love, trust, and empathy, (3) instrumental support—providing physical forms of help such as time or labour (e.g., helping to buy groceries), and (4) appraisal support—giving feedback to individuals which allows for self-evaluation (e.g., telling someone they did well) [13]. Countries with advanced health systems often possess significant professional support structures to aid mothers in their navigation of the perinatal period [14]. For example, comprehensive, uniform systems of care in some countries allow mothers from various backgrounds to receive low- (or even no-) cost guidance from medical doctors, midwives, paediatricians, and other medical health professionals. Notwithstanding a host of substantial beneficial (and often necessary) supports from these maternal health professionals, mothers sometimes report concerns—such as feeling judged or being stigmatised—when interacting with formal health care providers [15]. Moreover, studies on maternal treatment preferences have highlighted that formal health system support is often perceived by mothers as necessary but insufficient in assisting them to navigate the perinatal period [16,17]. Mothers have highlighted, for instance, that ongoing and consistent support from informal, non-professional sources, such as from families and friends, is essential throughout the postpartum period (see, e.g., [18]).

In response to mounting evidence that informal support systems bear a significant expression on maternal mental health, *peer-based social support programs* have become increasingly common as a set of non-professional, community-driven, and inclusive initiatives for supporting maternal mental health [19]. These programs differ from professional or clinical care because they have more focus on emotional support and are more readily available, and they differ from mothers’ or parents’ groups because they are less likely for parents to encounter competition or judgement [20,21]. Peer support programs are perceived by women as a low-stigma and effective option for the prevention and treatment of maternal mental illness [17,20], and may provide a number of benefits that align with maternal treatment preferences, including flexibility in how support is accessed—e.g., via WhatsApp, calls, online forums [22,23]. Another notable advantage of peer intervention programs is that they offer a platform for long-term friendships and ongoing, consistent, and non-judgmental support. Peer support interventions for mothers can take various forms; for example, they can incorporate trained or untrained supporters, telephone-based or in-home support, individual or group-based (i.e., one supporter assisting more than one mother) support, and they can be differentially focused (e.g., breastfeeding, postpartum depression, functioning and perinatal substance abuse) [16,24,25,26,27,28]. In general, research indicates that peer support interventions are effective in supporting outcomes such as improved breastfeeding practices, enhanced maternal self-efficacy, and reduced perinatal depression symptoms and substance use [27,29,30,31,32].

Despite evidence of benefits of peer support interventions for perinatal mental health, it is notable that individual mothers’ interactions with peer mentors can, at times, be undermining and stressful. Evidence shows, for example, that when grandmothers are highly involved and controlling as a parenting mentor, young mothers can experience low parenting efficacy [33], more parenting stress [34], and are less nurturing as mothers [35]. Peers may also have personalities that are incompatible with the mother, and mothers may also feel inadequate if they make comparisons with their peer [36]. This feeling of inadequacy can lead to suboptimal levels of support and result in poorer outcomes for mothers [37]. Thus, central to the effectiveness of peer support is the extent to which it is functionally adaptive, and in some cases, even good intentions from others may produce undesirable consequences. To maximise benefit from peer support, more research on peer social influences with respect to maternal outcomes is needed, and in particular, a focus is required on factors that may moderate the impact of peer support on mothers’ outcomes.

The importance of relationship processes is acutely felt among those who oversee programs in which peers with no pre-existing relationships (i.e., strangers) are paired [15,23]. Evidence shows that these types of programs—while often successful at a general level—produce highly variable results at individual and dyad levels (e.g., [38,39]). In a systematic review by Shilling and colleagues [39], the authors found inconsistent effects of peer support on psychological health (including anxiety and depression) and family function. They also reported that not all peer support relationships were successful in shaping positive outcomes for the recipients of support. To date, only a few studies have investigated factors influencing the strength of relationship between mentors and mentees in peer support programs for mothers [23,40]. For example, Shorey and Ng [23] found that having flexibility in mode of contact (e.g., phone calls, WhatsApp), being a listening ear to mentees, and helping mentees feel like they are not alone facilitated relationship development. By having a better understanding of the factors that influence peer support relationships, it may be possible to design programs that take those factors into consideration and improve the likelihood of successful peer mentoring relationships. To achieve this, we recruited participants from a feasibility study of The Mummy Buddy Program [38]. The Mummy Buddy Program was tailored to support primiparous mothers by providing them with a trained peer mentor who had successfully navigated the postpartum period within the prior five years. Program participants received support up to six months postpartum as pilot work had demonstrated that this was the period within which primiparous women experienced the greatest challenge [41]. Initial data indicated that the program was helpful in preventing increases in depressive symptoms and stress [38]. The aim of the present study was to investigate, through interviewing participants in The Mummy Buddy Program, factors that contribute to successful mentorship for primiparous women.

## 2. Materials and Methods

### 2.1. Philosophical Perspective

In line with concepts of constructive epistemology and ontological realism—which acknowledge the active role that the researcher and researched play in constructing knowledge about the social world—an interpretivist perspective was adopted [42] to gain insight into the factors that contribute to strong, close relationships in non-professional maternal social support programs. To achieve our research aims, and consistent with our epistemological stance, a qualitative design was employed. With the goal of providing useful, accurate, and reliable representation of accounts, we approached validity through concepts of “ontological plausibility (i.e., that various realities might exist), empirical adequacy (i.e., to have adequate data) and practical utility (i.e., applicability of findings)” ([43], pp. 22–24). In addition, a reflexive thematic analysis approach was adopted in alignment with our philosophical stance and research aims [44]. This was achieved through reflexive co-construction within the research, as exemplified by the interactions between the lead author and participants.

The research was conducted through a ‘Western’ English-speaking perspective as all researchers and participants were based in Australia. To prompt our interest in this research topic, we imagined that if we were first-time mothers (mentees) and peer mentors going through a peer support program, we were likely to have encounters which strengthened or weakened the mentor-mentee relationship. To understand the reality of others comprehensively it is important to be aware of our own differences from others [45]. In this regard, the lead author is male, of Chinese descent from Singapore, was a postgraduate student when collecting data, and does not have children. Several co-authors were parents at the time of data collection and analysis.

### 2.2. Procedures and Participants

#### 2.2.1. Participants

Email invitations were sent to participants in The Mummy Buddy Program at three time points: (1) after the new mother and mentor had met for the first time, (2) at three months postpartum for the new mother, and (3) at six months postpartum for the new mother. Recruitment took place between June 2018 to August 2019. A total of 36 phone interviews were conducted with 14 new mothers (Mage = 31.21, SD = 3.09) and 17 peer mentors (Mage = 33.31, SD = 3.11). Of the 14 new mothers, 11 identified as Caucasian, two as Asian, and one as Middle Eastern. Out of 17 mentors, 13 identified as Caucasian, two as Asian, and two as Eurasian. Eight mentors had one child, and nine mentors had two children. All participants had completed a high school diploma or obtained higher qualifications. Three new mothers (*n* = 6) and two peer mentors (*n* = 4) completed two interviews each, while all remaining participants completed only one interview (*n* = 26). Repeat interviews can reveal greater insight than one-off discussions [46,47]—in this instance, we offered participants the opportunity to provide additional information through a second interview as their relationship might have changed over time. Ethical approval for this study was granted by The University of Western Australia ethics review board. All participants provided informed consent at the beginning of the interviews.

#### 2.2.2. Mummy Buddy Program

The Mummy Buddy Program was a program designed to support first-time mothers from third trimester to six months postpartum. The program consisted of five components, (a) a three hour workshop for mentors on effective strategies for communicating and supporting mentees (e.g., empathy, self-compassion), (b) antenatal information sessions for mentees, which included parenting topics such as feeding strategies and babies’ sleep, (c) ‘Ready to Go’ planning sessions, in which mentor/mentee pairs were introduced and worked together to develop personalised support plans, (d) ongoing support provided to mentees by mentors, which was flexible and decided by individual pairs, and (e) a check-in phone call to mentees within the first six weeks postpartum by a non-profit, well-known community-based parenting support organisation (Ngala Parenting Services; www.ngala.com.au/, accessed on 12 October 2022) at four-to-six weeks postpartum. All workshops, antenatal information sessions, and ‘Ready to Go’ planning sessions were conducted face-to-face. A total of 45 pairs completed the program. Refer to [38] for the full program description.

#### 2.2.3. Data Collection

One-to-one phone interviews were conducted by the lead author using a semi-structured interview guide. The interview guide was co-developed by the lead author and two co-authors who are experienced in conducting semi-structured interviews. In line with our aims and epistemological stance, our interview guide was not based on a priori theoretical or analytical frameworks. Instead, the interview guide, which was the same for all time points, consisted of open-ended questions that tapped into their relationship with mentor/mentee (e.g., how did your engagement with your mentor/mentee change over time?), and what they felt were important factors that influenced their relationship (e.g., was your relationship with your mentor/mentee what you expected? And why do you think that?). Data collection was guided by the notion of data saturation—the point at which no new information were developed from the data, and additional interviews would unlikely yield novel information relating to the research question [48]. However, we conceived of the issue of saturation in pragmatic terms—we do not assume that our data ‘finalises’ the experiences of participants in the peer mentoring program, and it is entirely possible that further interviews would yield potentially novel findings. We chose to cease interviewing participants for pragmatic reasons, because interviews became repetitive, and because we had sufficient data to construct a clear narrative of results [49].

#### 2.2.4. Data Analysis

In accordance with the interpretivist nature of this study, a reflexive thematic analysis approach was adopted, following thematic analysis guidelines provided by Braun and Clarke [50]. All audio-recordings were transcribed verbatim, and the lead author familiarised himself with the responses. Within the whole data set, semantically similar responses were identified and initial codes were generated. Broader themes were then formed tentatively through clustering of related codes. While this process was primarily based on semantic similarities, latent meanings (i.e., coding of implicit concepts) of responses were also explored and included where relevant. Themes and sub-themes were created to better represent the data. A series of ‘critical friends’ meetings—a series of discussions between all authors to provide critical feedback on the interpretation and analysis of data—were conducted to refine the tentative themes [51,52]. These meetings provided an opportunity for themes (and sub-themes) to be re-named, re-ordered, and in some cases discarded. This process allowed for the best representation of data, which enabled insight into the factors that contribute to effective mentoring relationships for primiparous mothers.

## 3. Results

Four main themes were identified from the interviews: (1) expectations of relationship, which refers to the expectations held by mentors and mentees about how their relationship would work, (2) independence of peer mentor, which relates to having a mentor who was independent from the mentee and her immediate social network, (3) contact, which refers to the frequency of contact and factors that drive or reduce the amount of contact, and (4) similarities, which refers to similarities shared between mentor and mentee. Table 1 displays the themes with their definitions and includes some exemplar codes for each theme. Following the underlying philosophical assumptions of this research, frequencies are not reported [52]. Instead, relevance in addressing the research question and relationship to generalisable concepts within existing literature were given precedence. The same themes were shared by mentors and mentees.

### 3.1. Expectations of Relationship

Participants shared that they had certain expectations of, and from, their mentor or mentee, and that confirmation or disconfirmation of these expectations was important for the strength of relationships. For mentees, some expected mentors to be more proactive within the relationship. As one mentee expressed, “…she’s [mentor] quite laid back and relaxed. However, I guess maybe I would like to feel like she was sort of in charge a bit more with things, because I guess she’s the one that has gone through this before and it’s completely new for me…” Most mothers felt that expectation disconfirmation had a substantial negative impact on relationship quality; however, for some mentees, having their expectations unmet in their relationship with their mentor did not affect them too greatly. For example, one mentee indicated that “I’ve had lots of support and so it hasn’t bothered me, and I haven’t sort of missed it [mentor support] in a way. So, it hasn’t been an issue. But I think that was just what my expectation was”. As for mentors, some had expectations of being contacted more frequently by mentees as a resource or support. For example, one mentor indicated, “she [mentee] never really asked any questions or came to me with any issues or problems, which could have been because she didn’t have any. But I kind of felt like I was a little bit useless”. And another mentor mentioned, “I felt like a bit useless in a way because she didn’t really need my help to start with… But as her baby got older, the more questions she asked, the more I felt like I was helping. So, my expectations were met as her baby got older…”

Both mentors and mentees indicated that motivations to be polite and undemanding sometimes meant that expectations were not discussed. For example, one mentee said “Just that she [mentor] has been so generous with her time as it was. I didn’t want to think that she was incurring an expense on top of that”. And another mentee indicating “…only my desire not to impose too much on her because she’s busy with her two sons and she works as well”. Similarly, some mentors mentioned they did not want to seem ‘imposing’ or ‘nosey’ or make mentees feel like they were being ‘badgered’. Hence, they let their mentee ‘take the lead’ to determine how much contact was made between the pair. This lack of discussion meant that expectations were not met and had a negative impact on the relationship.

### 3.2. Independence of Peer Mentor

Mentors and mentees felt that it was important for mentors to be independent from the mentee and her immediate family and social circle. Mentors indicated that relatives, family, and friends carry with them ‘opinions and views on certain things’ which can lead to new mothers feeling ‘judged’ or ‘guarded’. One mentor shared, for example, “My friend is really close with her family and her hubby’s family but since having the baby they [family] were very opinionated and stepping on her toes a bit much, so she doesn’t really feel comfortable enough to go to them because she doesn’t really like what they have to say to begin with because it’s not really helpful”. And another mentor described, “…you are not being open, things don’t get discussed, things don’t get vented, you don’t talk about the really bad things, and their [family/friend] opinion will be put on you and they won’t see it any other way. So that’s why I think it’s better to have someone who doesn’t know you, who is totally neutral with their ideas”.

Mentees shared very similar sentiments to mentors, echoing the advantage of having an independent mentor. For example, one mentee commented, “your family can get in your face a little about things. But having an outsider is, like, non-biased”. Another mentee recalled, “So, if I was saying something that I would not be comfortable saying to someone who knew the person I wanted to talk about, like if I wanted to talk about even my partner or whatever, because she [mentor] was separate from my circle of friends it felt safer. I think that was the best thing that she gave me”.

One mentee compared her experience of having an independent mentor to her mothers’ group interactions, indicating that, “When you are talking to other people in your mothers group you have an expectation that it’s a reciprocal relationship, where they support you, but you also support them. Whereas in the mummy buddy program, I felt like I could just talk about my problems, and I didn’t have to think “oh, I will also have to be a support person for the person I am talking to”. I felt like I could talk about what I needed to talk about without thinking about what the other person needed from me”.

### 3.3. Contact

The frequency of contact between mentors and mentees varied between pairs. For some mentors, their mentees “didn’t seem as keen to keep in contact” and that they “didn’t talk as much”, but others had more contact with their mentee. For example, a mentor noted, “I was touching base with her once a week via text, asking how she was and if she wanted a call. So, we were touching base weekly, we would have a call every couple of weeks or so”. Additionally, another mentor shared, “Most of the time it is me checking in with her, every few days if I haven’t heard from her, asking how she is, how her baby is, how feeding’s going, how’s sleep is going and things like that”. The experience of mentees was similar—some had frequent contact with their mentors, for example, a mentee shared, “She [mentor] always just checks in, even via text. Even if we don’t catch up one week, she checks in. She even asks me questions, it’s been a really positive thing. I am so glad I did it”. Others had less contact; for example, another mentee mentioned, “I think things got pretty busy and she had her own stuff going on. So, I hadn’t heard from her in a little while… it’s just more that sort of ongoing contact. I haven’t had so much”.

Pairs who had more frequent contact seemed to enjoy better quality relationships. For example, one mentee indicated, “We caught up maybe four to five times since the introduction. Additionally, she just messaged me again and shared some personal news with me. She was a really good contact”. Additionally, another mentee said, “we have been chatting like every week, and sometimes even more often. We’ve also caught up a couple of times in person. It’s been really helpful. Just knowing that there is someone I can talk to”. Mentors shared similar experiences to mentees, with some mentors reaching out to mentees weekly which led to pairs ‘knowing each other a lot better’ and, for some, ‘became friends’. Conversely, less contact seemed to hinder relationship development. One mentee stated, “we haven’t spoken much, probably five times. The way we chose to communicate was using WhatsApp, which might have been our downfall…” Another mentee elaborated on a similar experience, “I would maybe have liked to have a closer relationship than what we ended up having but I think the big thing has been the time to do it [meet]. Going back to work a bit more has meant that she [mentor] isn’t as available and maybe me because I haven’t had as much contact with her so I haven’t formed as close relationship as I thought I might”. Mentors shared the same sentiments as mentees in relation to the importance of contact frequency for relationship quality. A few mentors shared that they had difficulty forming relationships with their mentee as they ‘didn’t really get a chance to know’ them as some mentees “didn’t want to see anybody for the first six weeks [postpartum]” or ‘wasn’t available very often.’ As one mentor noted, “Yeah, we probably keep in touch a lot less. Like I remembered, she was keen, she had a lot of questions, she said she was going to message me a lot and I said message me whenever you want but I think I probably got one text after that about a question and that was it”.

Two common factors seemed to influence the frequency of contact within each pair. Firstly, when important issues arose (e.g., relationship with partner or family, baby related challenges) there was often an increase in contact. For example, one mentor recalled, “yeah, once the baby arrived, I was definitely more of a sounding board for her and I heard from her a lot more”. A similar sentiment was shared by a mentee, who noted, “She has been really helpful throughout, but I think she was especially helpful during the first 6-weeks or even 3-months. Like those early stages, because you are trying to figure everything out”. Secondly, mentees who had a strong existing support network seemed to be less likely to contact their mentor. For example, a mentor indicated, “when her husband is around, I don’t have much involvement, which is fine. So, it’s only like a message here and there and still trying to keep in contact. But obviously she doesn’t need as much support”. A mentee also shared, “I have had quite a lot of support, so I have not had to use my mentor much. She usually gets in contact with me to touch base. Because I have friends who have kids between 1–5 years old, I tend to vent or talk about things with them when I need to. So, by the time I talk with my mentor, things would have been resolved or talked through”.

### 3.4. Similarities

The amount of similarity between mentors and mentees seemed to influence the strength of relationship within pairs. Mentors shared that being “different people” or having “different pregnancies” made it hard to “relate to” which resulted in difficulty building a strong relationship. One mentor said, “I think we had very different experiences and trying to relate my full-term baby versus her premi [premature baby] who was less than two kilograms at birth, and she was in hospital for a couple of weeks and got all these feeding [issues]… It’s like two completely different experiences”. Another mentor shared that even with similar experiences, she felt that her mentee and her had dissimilar personalities which seemed to impact their relationship development. This mentor said, “Our babies had similar stages and were quite clingy babies initially. I was really stressed out about it, but she seems to kind of go with the motion and I thought the person I would be paired up with would be more stressed out about it? I was surprised about that actually. She’s really chilled out for the whole thing”.

Mentees echoed parallel sentiments about how being similar to their mentors improved their relationship. As one mentee described, “Her kid ended up having a lot of things similar to my baby… a lot of things she had already experienced. So it was helpful to have another person who has experienced it to chat”. Another mentee commented, “We had a lot of things in common, like her kids are going to the school that I used to go to, and she lives in a similar area to where I grew up. She is not too much older than me and because I am a teacher, she asks me questions about her kids and school. That’s kind of how we developed a relationship”. Having shared experience, is not necessarily restricted to mothers themselves but can extend to their partner’s experiences. One particular mentee described that, “Seems like her [mentor] partner has gone through similar things as what my partner has so I think she has a lot to offer me that we can talk about those kind of things”.

Pairs who had something in common were more likely to lead to positive experiences. As one mentee said, “I think we clicked quite well. So, I felt quite comfortable just like chatting away the time we caught up last. I think we must have sat there for 2.5 h”. Additionally, another shared, “She’s been really wonderful and I feel like we are very well nest, very aligned and our thinking. She’s been a really great support”.

However, at least one mentee mentioned that she valued her mentor being different from her. This mentee mentioned, “I think she provides a different perspective to me. I have friends and family who are very similar to me. Whereas my mentor is probably a little bit different and she sees things from a different angle. So that’s probably the most useful thing”.

## 4. Discussion

Our aim in this study was to investigate factors that contribute to successful mentorship for primiparous women. The importance of this topic is underscored by previous research in which varied outcomes from peer support maternal health programs have been found (e.g., [39]). Data for the present study were obtained from ‘information rich’ mothers who had recently participated as mentees or mentors a specific peer support program for primiparous mothers. Our results highlighted that issues associated with expectations, contact, impendence of peer mentor, and similarity were especially important in shaping relationship quality between peers and perinatal women in peer support programs for maternal health. As such, this research sheds light on important considerations for designers and managers of peer support programs, especially those programs focused on mothers in the perinatal period.

The first theme in this study reflected a recognition from participants that confirmation of expectations between mentor and mentee was key in facilitating strong relationships. Adding on to existing literature, our study suggests that mentors and mentees may possess expectations in terms of frequency and type of contact, communication style, and type of support offered/welcomed. Data were suggestive of a possible moderator to the relationship between expectation confirmation/disconfirmation and relationship quality, with those having fewer alternative support options more greatly affected by expectation processes. Further research is required to verify this possibility. In terms of possible applications and implications related to programming efforts, given the broader—and sometimes unrealistic—expectations put on mothers [e.g., being able to breastfeed easily; 20], this theme highlights the importance of expectation-setting exercises at the commencement and throughout relationships in peer support programs. Expectation-setting exercises can help to keep expectations realistic, which in turn can prevent added difficulties and stress—as mentors and mentees will know what to expect throughout the relationship [53]. This theme of expectation confirmation suggests that key discussions about desires for frequency and nature of contact are important to discuss early on and often throughout the relationship. Additionally, given the often-evolving nature of women’s experiences in the perinatal period, it is possible that regular or periodic discussions about expectations might be appreciated by mothers [2]. It is plausible, for instance, that support-related expectations from a mentor/mentee might be different following childbirth than before childbirth, so separate discussions across these two periods may be merited.

Our second theme reflected a discussion from participants about the benefits of forming a relationship with someone with whom they have shared no previous history. Perinatal women highlighted the importance of having mentors who are not part of their existing social circle, as well-meaning family and friends might carry their own agendas, and may not be focused on the needs of the mother [54]. This theme is particularly interesting given that problems associated with stigma, judgment, and criticism have been given as reasons for perinatal women’s desire for non-professional (as opposed to professional) support [15,20,21]. Our data provides a more nuanced perspective on this issue—that certain types of non-professional support, especially from peers with whom no history has been shared, may be more likely to provide a less judgmental or critical context for perinatal women to share their experiences. This is possibly why mothers are more inclined to disclose sensitive topics with anonymous peers [55,56]. Correspondingly, it is concerning that some women perceive stigma, judgment, or criticism from existing friends and family; such feelings are likely to undermine healthy and adaptive functioning for mothers [57]. Future research is needed to investigate the process or factors that contribute to mothers’ perceptions of negative judgment or criticism, and whether interventions targeting these processes (in terms of mothers’ perceptions and/or behaviour from friends and family) can improve non-professional support. Additionally, research is recommended on decision making processes associated with mothers’ selection of sources of support, and how source selection processes may evolve throughout the perinatal period.

Our results reinforced findings from other studies demonstrating that contact frequency in a newly formed relationship is aligned with liking (e.g., [58]). For example, Reis and colleagues [59] found that more interaction led to increased familiarity and liking, even for unacquainted same-sex persons. Therefore, program designers should consider recommending regular meetings between pairs, and to facilitate contact through various mediums. Finally, as might be expected, primiparous mothers engaged with their mentors more when they experience difficulties (e.g., challenges with baby or navigating relationships). However, the level of increase in contact might be contingent on their existing support relationships. Some mentees seemed to prefer approaching their friends who have children in the first instance, rather than approaching their mentors. Again, this might relate to the nature of relationship that primiparous mothers have with their friends which determines whether they would contact their mentors. That said, during difficult periods mentors still indicated an increase in contact from mentees. This might reflect mentees not having the type of relationship where they can be open and share their issues with their support circle, and therefore choosing to contact their mentors instead [26]. Another reason could be because mentors are outside of their social circle and therefore provide a different perspective. Hence, mentees might value the utility of having a mentor as they provide strategies to deal with issues that they might not have thought about. More research is required to gain a better understanding of factors that influence new mother’s decision in who to engage with.

When primiparous mothers decide to engage with a mentor, then having similarities with their mentor can be an important driver for successful relationship. Similarities could be in terms of personality (e.g., laidback), background (e.g., went to the same school), style of parenting (e.g., authoritative), or experience (e.g., difficulty with breastfeeding). Perhaps having some form of similarity can provide a bridge that connects mentors with mentees, which supports the development of the relationship. Previous research has highlighted different areas of similarity, such as being in a similar geographical location, culture, employment status, marital status, age, and recency of childbirth, as contributors to peer relationship building [23,27]. Having similar experiences, however, seems to be one important aspect for successful relationships identified by many studies (e.g., [23,37,40]). Research on similarity-attraction suggests that surface-level (e.g., age, ethnicity) and deep-level (e.g., attitude, values) have impact on quality of social relation with peers and program adherence behaviours [60,61]. Thus, in addition to having shared experiences, future research should identify key similarities and other important factors that are essential for building relationship with peer supporters. The information can then be used to improve matching efforts in peer support programs.

## 5. Limitations

It is important to note that this study was conducted before the COVID-19 pandemic, and it is possible that some preferences for engaging with mentors might (for some new mothers) have since changed. Indeed, recent research has indicated that mothers experienced a decrease in social support during the pandemic [62], which might have increased their desire for more support [2], and therefore their potential engagement with a peer-mentoring program [63]. We also acknowledge that the majority of the participants in this program were of Caucasian descent, hence generalisation to other cultural backgrounds would need to be cautioned. Researchers should investigate whether similar findings can be replicated in other populations such as primiparous mothers in remote regions or First Nations people. Additionally, as the Mummy Buddy Program utilised independent pairing, it is possible that those who enrolled in the program were women who saw value in this approach. Whereas those who prefer friendship or family pairings, may not have enrolled in the program in the first instance. Finally, the interviewer was demographically dissimilar to interviewees (i.e., ethnicity, parental status, gender) which might have prevented more open discussion during interviews. However, this was unlikely as the interviewer had built strong rapport with interviewees due to his role in the Mummy Buddy Program.

## 6. Conclusions

In conclusion, results from this study provide some insight into the factors that can help to build successful peer support relationships. These factors include having the option of independent parties as supporters, having matching or setting expectations between mentors and mentees, encouraging more contact between mentors and mentees, and to match pairs based on some form of similarity (e.g., experience with baby). Importantly, future studies should investigate the decision mechanisms used by primiparous mothers in deciding when to engage with peer supporters. This can further support future intervention efforts and maximising the usage of peer supporters. Future studies can also look to successful peer mentorship programs, such as peer support programs for participants in the Supplemental Nutrition Program for Women, Infants, and Children [e.g., 32], to identify effective mentorship strategies.

## Figures and Tables

**Table 1 jcm-11-06009-t001:** Themes, definitions, and exemplar codes.

Theme	Definition	Example Codes
Expectations of relationship	Expectations held by mentors and mentees about how their relationship will work	I guess I wasn’t sure how much support my mentee would need. Like my mentee didn’t really need that much support, she handled it all really well. She didn’t seem to me like she needed a shoulder to cry on or big venting sessions or anything like I might have expected. (Mentor)
		Only my desire on not to impose too much on her because she’s busy with her 2 sons and she works as well? So, I would tend to just save up my questions and then we have the one phone conversation per week. (Mentee)
Independence of peer mentor	Having a mentor who is independent from the mentee and her immediate social network	I think friends are alright, but you sometimes need people who are out of the situation. If you are a relative, you are already in that ‘bubble’ you don’t really see things from another point of view sometimes. When you are so sleep deprived and you can’t think straight, somebody who you do not know or have a connection in that way, it’s a lot easier to hash things out, talk things over. You get so much more rational. (Mentor)
		I really liked the independent aspect to it. Completely outside of your friendship group and your family. Just so you can, I think it would open up people more, like they are more likely to discuss something more personal with someone who does not have that (social) connection with. Some people with family groups they might feel like they would be judged, if they are asking about certain things or their friends might have one view and they might have another. I think the independence is a great aspect of the program. (Mentee)
Contact	Frequency of contact and factors that drive or reduce the amount of contact	She gave birth a few days after meeting her, so I didn’t really get a chance to know her before everything started with her and she, not that we didn’t get along well, but she didn’t really want to see anybody for the first six weeks. She had her own rules in place, we communicated mostly via text messages and Facebook, and that was fine but she didn’t seem as keen to keep in contact unfortunately. But I check in with her every now and then and I have met up with her twice. (Mentor)
		I would maybe have liked to have a closer relationship than what we ended up having but I think the big thing has been the time to do it (to catch up). Going back to work a bit more has meant that she isn’t as available and maybe me cause I haven’t had as much contact with her so I haven’t formed as close relationship as I thought I might. (Mentee)
Similarities	Similarities between new mum and peer mentor (e.g., experience, character, beliefs)	So I’m wondering if there’s something more like (a criteria) based on ages, family situation or experiences, like mums who are planning to go back to work quite quickly, maybe like a different way to match people up, so you get a more similar… Like I’m a stay-at-home mum and I have been for 5 years, and I have people who are at six months and they are desperate to drop their kids at day care and go back to work. So we are very different, we have different personalities in that way. (Mentor)
		Her (mentor) kid ended up having a lot of things similar to my baby. So my baby was born with one clubbed foot. So he had a lot of treatment, like a cast and a brace on his feet, and her kid has had hip dysplasia… a lot of things she had already experienced so it was helpful to have another person who has experienced it to chat through. (Mentee)

## Data Availability

Not applicable.

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
