# Peer review of "Strengthening Peer Mentoring Relationships for New Mothers: A Qualitative Analysis"

_jcm, 2022, doi:10.3390/jcm11206009_

Round 1

Reviewer 1 Report

This is a well-written report of qualitative findings regarding factors associated with strong peer mentoring relationships for new mothers. It will be a helpful contribution to the field for peer-delivered services and to inform future research. I had a few minor questions/suggestions for improvement or clarification: 

- The sample size was quite small and I don't see reporting of demographics - how homogenous was the sample? how was sample recruited for Mummy Buddy Program, and what % of those participants provided these qualitative interviews? 

- It seems a bit questionable to reach data saturation with such a small sample - good that you justify for pragmatic reasons; was it saturation, or pragmatic reasons? 

- if you did 3 interviews for each participant over time, did you see anything about change in themes over time? 

- Were there differences in themes between mentors and mentees? If not, I'd report that...
